# Assessing the Physiochemical Parameters and Reduction Efficiency from Two Typical Wastewater Treatment Plants in the Vhembe District in South Africa

**DOI:** 10.3390/ijerph22060856

**Published:** 2025-05-30

**Authors:** Leonard Owino Kachienga, Thendo Ndou, Mpumelelo Casper Rikhotso, Afsatou Ndama Traore, Natasha Potgieter

**Affiliations:** Department of Biochemistry and Microbiology, Faculty of Sciences, Engineering & Agriculture, University of Venda, P/bag x5050, Thohoyandou 0950, South Africa; 21002581@mvula.univen.ac.za (T.N.); mputso@yahoo.com (M.C.R.); afsatou.traore@univen.ac.za (A.N.T.); natasha.potgieter@univen.ac.za (N.P.)

**Keywords:** physicochemical, wastewater effluent, sewage treatment, pollution, reduction percentage, guideline limit

## Abstract

The primary sources of widespread pollution in most river catchments are improperly treated final effluents from various wastewater treatment plants (WWTPs), affecting the physicochemical characteristics of the receiving water bodies. Wastewater discharge must be monitored regularly to ensure compliance with national and municipal water quality regulatory/standard restrictions. This study monitored the physicochemical parameters of two typical WWTPs (WWTP A = a peri-urban plant and WWTP B = a rural plant) for 5 months. The physicochemical parameters that were assessed included pH, temperature, total dissolved solids (TDSs), turbidity, chemical oxygen demand (COD), alkalinity, dissolved oxygen (DO), free chlorine, chloride, sulphate, phosphate, ammonium, and electrical conductivity (EC). The evaluation yielded the following results: temperature (14 to 21 °C and 14 to 23 °C), pH (7.2–8.2 and 7.3–8.4), EC (90–800 μs/cm and 80–750 μs/cm), TDSs (65–440 and 55–410 mg/L), alkalinity (2.6–20.9 mg/L), nitrate (0.24–26.5 mg/L), nitrite (0.01–90 mg/L), phosphate (0.0–18.0 mg/L and 0.0–21 mg/L), ammonia (0.2–75 mg/L and 0.8–70 mg/L), sulphate (0.0–18.0 mg/L and 0.0–21 mg/L), chloride (5.0–22.0 mg/L and 2.0–25 mg/L), COD (6.0–710 mg/L and 7.0–800 mg/L), and turbidity (0.4–150 NTU and 1.8–130 NTU) for wastewater treatment A and B, respectively. The results also showed that temperature, pH, TDSs, nitrite/nitrate, chloride, turbidity, alkalinity, sulphate, and free chlorine were among the parameters in the final effluent discharged that met the set guidelines. In contrast, parameters such as COD, EC, phosphate, and ammonia did not meet the guideline values for most of the sampling period for both WWTPs. Furthermore, this study found that WWTPs reduced nitrate, sulphate, phosphate, and COD pollutants by more than 90% while maintaining 60% alkalinity. Temperature, pH, TDSs, EC, Cl^−^, and other parameters were less than 40% for WWTP A and roughly less than 50% for WWTP B.

## 1. Introduction

Water is one of the most vital and significant resources necessary for life, yet it is a global challenge that compromises the amount or quality of the available water resources [1,2]. About 2 billion people worldwide lack access to safe drinking water and adequate sanitation, according to Hokkanen et al. [3] and Pereira and Marques [2], respectively. Due to anthropogenic activities (pollution, mining, recreation, and agriculture), population growth, and ageing infrastructure, inadequate freshwater remains a significant problem in developing and some developed nations [4,5,6,7]. It is crucial to treat wastewater before it is released back into water resources or catchments or for reuse, since freshwater pollution can have disastrous consequences, particularly in water-scarce nations like South Africa (SA) [8,9].

To protect these accessible freshwater resources and aquatic life, wastewater treatment facilities are crucial [10]. Therefore, it is necessary to consider integrating low-cost technology policies that will assist in improving the efficiency of wastewater treatment by reducing energy consumption with a low carbon footprint and further diminishing the formation of waste in water and the environment [11]. There are more negative effects on human health and the environment when untreated or partially treated wastewater is released into water sources and their surroundings by malfunctioning wastewater treatment plants [10,12]. Furthermore, according to Naidoo and Olaniran [13], additional challenges to stress water resources and their sustainability stem from various categories of complex and highly toxic chemical and microbiological constituents of the pollutants discharged into these resources. Bănăduc et al. [6] also reported that the devastation of aquatic species has a detrimental effect on the aquatic ecosystem that sustains recreational activities as well as disrupts the current food chain.

The key factors influencing water quality are its physicochemical properties and the level of contamination [10]. Abdel-Raouf et al. [14] and Mbalassa et al. [15] further suggested that these qualities of optimal freshwater can also assist in determining the best circumstances for aquatic ecology and create effective conservation and management plans. Most water resources are contaminated by different sources of pollutants [16]. These include water-soluble inorganic pollutants like caustics, salts, acids, and toxic metals (ammonium salts, nitrates, phosphates), oxygen-demanding waste (biodegradable materials like plant residues and animal manure) that is naturally or artificially added to the water, and pathogen-causing agents (bacteria, viruses, protozoa, parasitic worms, and bacteria that cause waterborne diseases). The main nutrients that cause cyanobacteria and algae to thrive and cause eutrophication are phosphates and nitrates [17,18].

The capacity and effectiveness of the receiving wastewater treatment facilities are established by the ultimate quality of the effluent released into the freshwater sources/catchments [19]. The primary responsibility of wastewater treatment facilities is to reduce both organic and inorganic waste by either lowering nutrient loads or raising the dissolved oxygen (DO) in the final treated influents. It also protects human health by deactivating disease-causing microorganisms [20,21].

The technologies used by different WWTPs include membrane filtration, oxidation ponds, activated sludge, and biofilters, amongst others. The type of wastewater that needs to be treated from different sources, such as industrial, pharmaceutical, agricultural, mining, or residential, and economic and environmental factors influence the decision. Several cases such as rapid industrialisation, widespread poverty, and climate change have contributed significantly to the recent increase in the reuse/reclamation of final effluents from treated wastewater [22,23]. Similarly, fast-developing nations like South Africa, which is also semi-arid, require adequate tools to assist in monitoring the overuse of water stored in reservoirs, dams, or groundwater sources. This can only be achieved if wastewater is recovered or reused [24]. Few studies have been carried out in rural or peri-urban settings such as the Vhembe District, with the bulk of studies evaluating the physicochemical analysis quality of the final effluents from various treatment plants mostly focused on urban effluents and their impacts on their adjacent river sources in developing countries like South Africa. Therefore, it is imperative to perform these kinds of studies with expanded parameters (16) influencing water quality, especially in far-flank areas of Vhembe District, which will have a direct impact on the rural communities that are mostly bearing the brunt of inadequate access to clean water for domestic usage, water pollution, and waterborne ill health. Thus, the primary objective of this study was to evaluate the physicochemical parameters influencing the final effluents discharged from two typical WWTPs located in the Vhembe District’s rural and peri-urban areas and their efficiencies.

## 2. Methods and Materials

### 2.1. Study Area Description

The Thohoyandou and Malamulele regions are home to WWTPs A and B, which were taken into consideration for this study. These WWTPs are located in the Vhembe District of the South African province of Limpopo and are governed by the municipalities of Collins Chabane and Thulamela, respectively. WWTP A’s coordinates are 30°28′28″ E and 23°0′13″ S. According to Edokpayi et al. [25], this plant is regarded as peri-urban and receives wastewater from the villages and peri-urban areas surrounding Thohoyandou, Shayandima, and Manini. It also receives wastewater from light industrial effluent, the University of Venda, medical facilities and clinics, and agricultural enterprises. According to the 2023 Department of Water and Sanitation, Green Drop Report [26], WWTP A’s actual operational capacity is 116.2%, whereas its design capacity is 12,000 m^3^/d. Notably, the Mvudi River receives the final processed effluent from WWTP A. Notably, the Mvudi River receives the final processed effluent from the Thohoyandou wastewater treatment plant. WWTP B is situated along the Mandzoro River in Malamulele, section C, at −23.0635° latitude and 30.7432° longitude. It takes waste effluents from Malamulele Centre and the nearby communities and is regarded as a rural WWTP. This plant has an operational capacity of 166.7% and a design capacity of 3000 m^3^/d. The technological flow of WWTPs A and B was as follows: Pre-treatment of the incoming wastewater influent occurred using a manual screening chamber, and then it was channelled into the trickling/biofilters. The water was then passed through oxidation ponds for treatment, and the final effluent was disinfected using chlorine dosage before being discharged into the adjacent rivers (Appendix A).

### 2.2. Collection of Water Samples

The samples were taken from influents and effluents for WWTPs. The samples were also taken from the upstream and downstream rivers (Mvudi and Mandzoro) where final effluents are ultimately discharged (Appendix A). This was performed once a month between June 2024 and October 2024. Sterile polyethene bottles were used to collect the samples, which were then transported to the University of Venda One Health laboratory on ice and processed immediately. 

### 2.3. Physicochemical Analysis of Samples

Following the manufacturer’s instructions, all metres and equipment used for parameter analysis were calibrated and inspected. Temperature, pH, electrical conductivity (EC), total dissolved solids (TDSs), dissolved oxygen (DO), chemical oxygen demand (COD), alkalinity, and the concentrations of nitrate, nitrite, phosphate, sulphate, chloride, ammonium, turbidity, and total/free chlorine were amongst the physicochemical parameters that were examined. A Hanna HI98194 multi-parameter reader was used to analyse temperature, pH, electrical conductivity (EC), total dissolved solids (TDSs), and dissolved oxygen (DO). A Hanna H1801 spectrophotometer iris parameter was used to analyse alkalinity, the concentrations of nitrate, nitrite, phosphate, sulphate, chloride, and ammonium, and total/free chlorine. Both instruments were manufactured by Hanna Instruments in the USA. Using a Lovibond TB 211 turbidimeter (Lovibond, IR, Germany), the samples’ turbidity was assessed (Appendix A). Lastly, Hanna Instruments in the USA used a thermo reactor Model TR 300 to digest chemical oxygen demand (COD) samples. The concentrations were then measured using an iris parameter on a Hanna H1801 spectrophotometer (Appendix A). All the analyses were performed in duplicate.

### 2.4. Wastewater Treatment Efficiency for WWTPs A and B

Efficiency was determined using Equation (1) [27] (1)$$\%\mathrm{Reduction}\;=\;\frac{C_{i}-C_{f}}{C_{i}}$$
where *C_i_* and *C_f_* are the concentration of waste matter in influent and effluent from WWTPs, respectively.

## 3. Results and Discussion

### 3.1. Physicochemical Parameters of the Samples

Table 1 and Table 2 represent the findings of the physicochemical characteristics of the wastewater samples that were taken from WWTPs A and B, respectively. During the study period, WWTPs A and B had temperature profiles ranging from 14 to 21 °C and 14 to 23 °C, respectively. For every month, they both adhered to the established discharge effluent limitations. For every WWTP, September through October had the lowest temperatures (Table 1 and Table 2). The predominant rainfall that was seen during the sampling period was the reason for the lower temperatures. To prevent overflow from excessive rainfall, the effluents in the current oxidation ponds might have had a shorter holding duration [10]. The presence of industrial effluents treated by WWTP A may decrease the temperature more than WWTP B, which exclusively receives influents from residential houses and nearby villages, according to Olabode et al. [5]. The variations result from cooling processes that cause thermal pollution from effluents produced by industrial sources [28]. Iram et al. [29] and Olabode et al. [5] similarly found a temperature variation of 18.1 and 28.5 °C from three distinct sample times, which they also attributed to seasonal changes. 

During the study period, the pH values of WWTPs A and B ranged between 7.2 and 8.2 and 7.3 and 8.4 (Table 1 and Table 2), respectively, and both met the DWAF [30] guideline of 5.5–9.5 for the effluents released into receiving waterbodies. These results were comparable to those of Tikariha and Sahu [31], who reported wastewater effluent from a dairy farm in the Bilaspur belts of Chhattisgarh of India showing a similar tendency of a pH range between 6.0 and 8.0. According to Agoro et al. [10], the availability and adaptability of micronutrients are crucial for determining the acid–base level of water. The solubility of various chemical contaminants, other significant components in surface water, and the harmful impacts on aquatic life are all influenced by changes in pH. This also negatively affects the whole water ecosystem [30]. 

The values of DO for both wastewater treatment plants were extremely low and below the guideline set by the DWAF [30] of 6.5–8.0 ppm, except for October, which had high DO counts and was above the guideline limit (Table 1 and Table 2). The ability of most waterbodies to self-purify is a critical component in directing both physical and biological processes in a particular water body and DO is crucial in assessing the degree of contamination by any organic pollutant [10,32]. Because there is little or no oxygen available to them, low DO typically disrupts the existence of aquatic animals, which harms them through disease exposure, migration trends, reproductive behaviour, swimming capacity, feeding patterns, and other factors [5,10,33]. The authors also believe that as DO declines, so do the inorganic chemicals found in the nearby industries’ wastewater.

During the study period, WWTPs A and B had total dissolved solid (TDS) levels ranging from 65 to 440 and 55 to 410 mg/L, respectively (Table 1 and Table 2). Despite variations over the months, they both met the DWAF-established guideline limit (<450 mg/L). These results contradict the findings of Mahananda et al. [34], who proposed that most inorganic compounds are the main constituents of TDSs and that wastewater from industrial facilities is the source of higher values. Additionally, according to Qadir and Chippa [35], the amount of TDSs in water has a more significant effect on the water’s density, which directly affects aquatic species’ osmoregulation. 

For WWTPs A and B, the electrical conductivity (EC) measured during the study period ranged from 80 to 750 μs/cm and 90 to 800 μs/cm, respectively. Throughout the months, they were all above the DWAF-established standard threshold of less than 70 μs/cm (Table 1 and Table 2). August through October produced the greatest value for both WWTPs. In September (WWTP A) and October (WWTP B), the nearby rivers similarly had EC values of more than 600 μs/cm (Table 1 and Table 2). The dissolution of ions from the surroundings or decayed dried plants was the cause of the rising trend. Employing ionisation processes, the dissolved inorganic matter also contributes to elevated EC levels [10]. Higher chlorine concentrations also encourage higher levels of EC in water [36]. The rise in EC, particularly in the WWTPs’ neighbouring rivers, particularly around September and October, was also comparable to the findings of studies by Singh et al. [37] and Ewemoje and Ihuoma [38], who observed TDSs of >600 μs/cm, respectively, in a study that assessed the physicochemical parameters of black water discharge on River Zik in Nigeria and characterised antimicrobial India. A useful measure of salinity with total salt content is the EC of surface water. According to Odjadjare and Okoh [39], aquatic species and animals that use water straight from the discharged rivers suffer from impaired osmoregulatory systems when exposed to high amounts of TDSs. According to Chaurasia and Pandey [40], the concentration of EC at any given period is directly correlated with the TDSs and salinity of the water.

**Table 1 ijerph-22-00856-t001:** Physicochemical data for WWTP A.

Month	Sampling Point	Temp °C	pH	DO (ppm)	TDSs (mg/L)	EC (μs/cm)	Turbidity (NTU)	Ammonia (mg/L)	Nitrate (mg/L)	Nitrite (mg/L)	Phosphate (mg/L)	Sulphates (mg/L)	RC (mg/L)	TC (mg/L)	Chloride (mg/L)	Alkalinity (mg/L)	COD (mg/L)
**June**	IN	20.75	7.24	0.00	314.5	491	19.5	67.2	12.50	12.5	17.4	39.0	0.22	0.01	22.0	342.5	638
	FE	20.2	7.40	2.16	197.8	309	13.2	30.00	2.00	2.00	11.15	31.0	0.04	0.05	20.0	206.5	63.0
	UP	17	7.50	5.25	57.35	89.6	5.60	1.83	0.00	0.00	0.80	16.0	0.18	0.11	7.00	173	11.0
	DS	17.1	7.48	4.70	89.3	138.4	6.73	3.40	0.00	0.00	2.70	2.00	0.05	0.03	18.0	705	20.0
**July**	IN	18.8	7.53	3.06	371.5	493	53.55	72.0	11.85	25.5	15.1	42.5	0.05	0.06	20.0	285	410
	FE	18.23	7.53	3.15	278	309	12.05	34.65	2.75	4.00	11.0	36.0	0.11	0.03	19.6	215	99.0
	UP	20.03	7.42	4.48	286.5	89.6	1.57	0.25	5.85	0.00	0.04	1.50	0.08	0.07	6.00	200	16.5
	DS	21.3	7.61	0.00	289	138.9	2.08	3.60	5.15	0.00	0.95	2.00	0.06	0.11	16.8	214	13.5
**August**	IN	18.16	7.77	0.00	313.5	627	1.32	38.6	5.70	21.5	15.5	54.0	0.03	0.06	20.0	310	508
	FE	20.39	7.97	0.93	352	703.5	3.9	39.2	0.00	5.50	13.15	37.5	0.065	0.25	18.5	232	196
	UP	16.29	7.78	1.94	208.5	419	0.99	6.40	6.25	0.00	0.00	0.50	0.00	0.00	5.8	774	18.0
	DS	19.0	7.75	1.59	144	286.5	0.47	1.25	5.25	0.00	13.1	2.00	0.00	0.00	9.95	183.5	7.00
**September**	IN	19.44	8.15	0.00	443.5	887.5	13.0	47.8	30.0	0.00	15.7	74.5	0.61	0.68	20.0	316.5	601.5
	FE	13.31	7.78	18.86	339	678	5.85	35.7	0.00	3.00	0.00	0.00	0.15	0.00	20.0	238.5	139.5
	UP	15.55	7.80	2.45	307	615.5	6.19	10.6	6.95	0.00	0.65	21.5	0.12	0.21	20.0	184	54.5
	DS	13.99	7.94	2.38	320	641.5	7.19	1.75	4.55	0.00	13.7	46.5	0.03	0.02	16.5	204.5	6.00
**October**	IN	14.75	7.60	1.3	382.5	765.5	149	46.5	30.4	0.00	22.0	66.0	1.48	0.89	14.5	316	702.5
	FE	15.32	7.49	16.9	254	508	10.79	23.5	0.00	5.50	0.00	0.00	0.59	0.49	17.65	128.5	144
	UP	13.99	7.62	20.25	68.0	136	50.35	9.50	16.9	0.00	0.00	27.5	0.00	0.00	5.55	42.5	22.5
	DS	14.01	7.55	21.5	121.5	243	24.05	2.05	15.7	0.00	12.0	36.0	0.04	0.06	13.2	66.5	27.5
DWAF [30,41] guidelines		35 °C	5.5–9.5	6.5–8.0 ppm	450 mg/L	70 μs/cm	10–20 NTU for river water containing sediment	3 mg/L	15.0 mg/L	15.0 mg/L	10 mg/L	200 mg/L	0.25 mg/L	0.25 mg/L	100 mg/L	20–200 mg/L	75 mg/L

Key: DO = dissolved oxygen, TDSs = total dissolved solvents, EC = electrical conductivity, RC = free chlorine, TC = total chlorine, IN = influent, FE = final effluent, UP = upstream, DS = downstream.

**Table 2 ijerph-22-00856-t002:** Physicochemical data for WWTP B.

Month	Sampling Point	Temp °C	pH	DO (ppm)	TDSs (mg/L)	EC (uS/cm)	Turbidity (NTU)	Ammonia (mg/L)	Nitrate (mg/L)	Nitrite (mg/L)	Phosphate (mg/L)	Sulphates (mg/L)	RC (mg/L)	TC (mg/L)	Chloride (mg/L)	Alkalinity (mg/L)	COD (mg/L)
**June**	IN	21.9	7.64	0.00	365	570	15.3	68.2	26.5	26.5	10.8	19.5	0.01	0.00	23.6	130	713
	FE	18.8	7.54	2.15	200	312	14.3	9.80	1.00	1.00	9.80	38.5	0.25	0.18	21.2	165	48.0
	UP	18.9	7.45	5.25	56.9	88.3	5.88	1.60	0.00	0.00	2.05	3.00	0.23	0.03	10.1	59.5	7.00
	DS	18.5	7.44	4.70	83.5	129	6.87	1.22	0.00	0.00	2.30	4.60	0.25	0.29	19.5	16.6	26
**July**	IN	20.9	7.36	0.00	408	580	28.1	56.3	30.0	59.1	12.3	54.0	0.66	0.69	25.6	241	759.5
	FE	20.6	7.47	1.06	320	312	23.0	58.3	9.15	13.0	15.03	45.5	0.50	5.00	22.5	182	121.5
	UP	21.8	7.31	6.67	81.5	88.3	5.55	1.50	5.50	7.05	0.00	1.00	0.24	0.06	10.1	68.0	4.00
	DS	22.6	7.50	5.25	130	129	8.17	8.17	6.45	4.00	2.04	8.00	0.69	0.41	17.8	91.0	23.0
**August**	IN	22.8	7.76	0.00	360	720	2.17	40.0	30.0	4.00	26.2	68.5	0.78	0.85	21.8	256	797
	FE	21.1	7.94	0.00	220	546	4.79	24.3	0.00	25.7	11.2	42.0	0.00	0.00	19.7	168	130
	UP	20.0	7.78	1.67	72.0	144	5.89	25.3	3.20	0.00	0.00	1.00	0.00	0.00	5.10	38.5	9.00
	DS	21.7	7.67	2.42	56.5	114	5.58	0.85	5.05	0.00	12.0	2.00	0.00	0.00	11.9	43.5	57.0
**September**	IN	14.8	7.78	0.00	357	714	89.1	44.9	27.8	0.00	21.4	56.5	2.00	0.83	20.0	315	800
	FE	15.3	7.77	18.9	123	446	44.6	18.8	0.00	0.00	0.00	0.00	0.20	0.10	20.0	129	84.5
	UP	14.0	7.63	2.45	76.5	153	1.91	5.75	3.60	0.00	0.00	27.0	0.00	0.04	20.0	41.5	22.5
	DS	14.4	7.65	2.38	96.0	192	2.01	1.35	4.40	0.00	0.50	4.00	0.04	0.07	17.7	66.0	18.0
**October**	IN	19.4	7.78	0.00	356	713	125	50.4	32.5	0.00	19.0	82.5	1.61	0.64	20.0	317.5	629
	FE	13.3	8.35	28.4	354	708	30.9	35.5	7.63	8.50	0.00	0.00	4.38	5.09	14.2	242	138
	UP	15.6	7.86	29.1	315	629	14.5	12.0	13.3	0.00	0.18	28.0	0.14	0.18	2.95	187	58.0
	DS	14.0	7.81	22.6	234	467	5.13	3.15	22.5	0.00	16.75	47.0	1.18	1.21	12.15	212.5	56.5
DWAF [30,41] guidelines		35 °C	5.5–9.5	6.5–8.0 ppm	450 mg/L	70 μs/cm	10–20 NTU for river water containing sediment	3 mg/L	15 mg/L	15 mg/L	10 mg/L	200 mg/L	0.25 mg/L	0.25 mg/L	100 mg/L	20–200 mg/L	75.0 mg/L

**Key:** DO = dissolved oxygen, TDSs = total dissolved solvents, EC = electrical conductivity, RC = free chlorine, TC = total chlorine, IN = influent, FE = final effluent, UP = upstream, DS = downstream.

During the study period, the turbidity values for WWTPs A and B ranged between 0.40 and 150 NTU and 1.8 and 130 NTU, respectively. These values were within the acceptable guidelines for the DWAF [41] of 10–20 NTU for river water containing sediments (Table 1 and Table 2). As a result, the downstream/upstream and the final effluent from the wastewater released in the catchments of nearby rivers are fit for residential use. Additionally, Osode and Okoh [32] noted that high turbidity typically interferes with wastewater treatment stages, including coagulation and sedimentation procedures, raising the cost. In the tertiary treatment of wastewater, high turbidity also causes trihalomethane to develop during the chlorination process [10,32]. This also affects aquatic microbes’ survival further impacted by the water’s high turbidity, particularly when biofilms or highly suspended particles are present [42].

For WWTPs A and B, the ammonia concentrations ranged from 0.20 to 75 mg/L and 0.8 to 70 mg/L, respectively. Both facilities’ ultimate effluent concentrations exceeded the permissible DWAF [30] guideline limit of less than 6 mg/L (Table 1 and Table 2). During the study period for both facilities, the months of September and October, respectively, showed an elevated level of ammonia upstream of the discharging rivers (Table 1 and Table 2). Rain-washed fertiliser runoff, particularly during September and October, was the cause of the high ammonia levels, particularly upstream of the rivers [43]. The authors further suggested the inefficiency of the plant’s anaerobic digesters, which aid in the nitrification processes, or industrial waste or seeping sewage from the area around WWTPs.

Except in October, when the upstream and downstream of the nearby rivers reported higher values that were beyond the acceptable set limits, the concentrations of both nitrite and nitrates were within the acceptable ranges (Table 1 and Table 2). Regardless of the concentration level, nitrate is the best inducer of eutrophication, which disrupts biodiversity, produces an unpleasant water odour, and lowers the recreational value of water catchments [44,45]. According to Majumder et al. [46], livestock and agricultural fertilisers are the most frequent sources of nitrates in water bodies, and their presence in the water is a sign of bacterial development or activity. Aquatic organisms typically develop methemoglobinemia in response to elevated nitrite levels in water bodies [47].

During the study period, WWTPs A and B had phosphate concentration levels of 0–18.0 mg/L and 0–21 mg/L, respectively. The final effluent’s concentration exceeded the DWAF [30] guideline requirement of >10 mg/L (Table 1 and Table 2). Agoro et al. [10] provided phosphate data that were below the permissible standard limit in a typical three-rural WWTP, which contrasted with these results. According to Shrestha et al. [48], the overuse of fertilisers for agricultural purposes and effluents containing detergents from home wastewater and meat processing businesses contribute to high phosphate levels in water bodies. The inability of phosphate-reducing bacteria to absorb phosphates during the outdated and inefficient biofiltering processes is another factor contributing to the elevated phosphate content observed in the final effluents of both WWTPs. High phosphate levels promote eutrophication in aquatic environments, increasing oxygen demand and potentially leading to several undesirable ecological issues [10,49,50].

During the experimental period, WWTPs A and B achieved sulphate concentration levels of 0–18.0 mg/L and 0–21 mg/L, respectively, which met the predetermined threshold of less than 200 mg/L (Table 1 and Table 2). Sulphur and its ions, which naturally occur through leaching from gypsum and other common minerals, break down into different components to create sulphate in water [10,22]. According to Njoku et al. [51], the detergents and soaps used by residents living near the WWTP facilities are typical sources of sulphates.

The DWAF [30] states that free chlorine levels should not exceed 0.25 mg/L. For WWTPs A and B, the range of free chlorine measured was 0.00 mg/L to 1.48 mg/L and 0 mg/L to 4.38 mg/L (Table 1 and Table 2). It was determined that these results complied with the established DWAF general limits. This investigation was comparable to one conducted by Karikari and Ampofo [52], who found that infections can survive via the distribution system when free chlorine concentrations are low. Therefore, it is critical to regularly control and monitor the quantities of free chlorine in WWTPs. In July, WWTP B reported a high chlorine overdose, which implied that all the microbiological pollutants in the water body would be eliminated. However, this could also harm the aquatic species [53]. Chloramines are created when more chlorine is used, which might harm fish and cause other possible health problems [53]. Giardia and Cryptosporidium protozoa cannot be destroyed by the typical chlorine residual contact period [54]. Similar results were also reported by Agoro et al. [10], who evaluated the physicochemical characteristics of three typical wastewater treatment plants in South Africa. The values ranged from 0 to 0.22 mg/L, 0 to 0.17 mg/L, and 0 to 0.48 mg/L.

The chloride (Cl^−^) concentration levels obtained were 5.0–22.0 mg/L and 2.0–25 mg/L for WWTPs A and B, respectively (Table 1 and Table 2), over the research period, and these values satisfied the specified criterion of <100 mg/L (Table 1 and Table 2). Agoro et al. [10] reported 3.25–224 mg/L, while Iram et al. [29] reported 127.72–396.16 mg/L, which were extremely high for the final effluents from wastewater treatment plants in the Eastern Cape, South Africa, and Nullah Lai and Kohe-Noor textile in Pakistan, respectively. These results contrasted with those of these studies.

Because sodium chloride (NaCl) is a common dietary component that is excreted in wastewater through urine or faeces, its presence is directly proportional to the content of chloride [10,55]. Additionally, Chigor et al. [55] noted that when the quantity of chloride in water exceeds 250 mg/L, the taste of the water is affected, which can disrupt aquatic life and render the water unsuitable for agricultural use. These findings, however, indicate that the ultimate effluent released into the nearby rivers is suitable for aquatic life or agricultural use. According to Aniyikaye et al. [4], excessive levels of chloride in water also have an impact on aquatic organisms’ reproductive habits and the sustainability of their ecological food supplies.

During the research period, the alkalinity profiles at WWTPs A and B ranged from 40 mg/L to 780 mg/L and 40 mg/L to 320 mg/L, respectively (Table 1 and Table 2). In contrast to upstream, low alkalinity levels were found downstream from the WWTP B effluents plant. Throughout the investigation, both WWTPs A and B had relatively high alkalinity levels, which fell short of the established threshold of 20–200 mg/L (Table 1 and Table 2). An increase in the alkaline character of the effluents due to anaerobic decomposition may be caused by the industrial effluents released at the plant and the extended retention of the effluents in oxidation ponds. The presence of calcium, salt, and potassium compounds such as bicarbonate, carbonate, and hydroxide may cause variations in water alkalinity. These findings, however, were at odds with earlier observations of water samples with lower alkalinity levels, particularly those between 20 and 110 mg/L and 110 and 149 mg/L, as documented by Patil and Platil [56]. On the other hand, Patil and Platil (2010) found that alkalinity levels were much greater, ranging from 170 to 870 mg/L and 210 to 910 mg/L, respectively.

The established guideline for COD is less than 75 mg/L, according to the DWAF [30]. During the experimental period, WWTPs A and B had chemical oxygen demand (COD) ranges of 6–710 mg/L and 7–800 mg/L, respectively (Table 1 and Table 2). While the upstream and downstream samples conformed throughout the research, the released effluents in both WWTPs significantly exceeded the established guidelines. Chigor et al. [55] stated that COD is a potent oxidant that is necessary for the breakdown of both organic and inorganic materials. Continuous effluent discharge with high COD levels will negatively impact the receiving water body, and it is also helpful in defining hazardous circumstances and the presence of biologically resistant chemicals [57]. In the end, this will affect the health of the river catchment, particularly downstream, and the quality of the freshwater ecosystem [58]. The drinking water obtained from the treatment plants based on efficiency and costs may be impacted [57]. Similar results, ranging from 23.70 to 898.58 mg/L, were reported by Olabode et al. (2020) [5] in research that evaluated the physicochemical characteristics of two wastewater treatment plants in Cape Town based on water quality.

### 3.2. Efficiency of the Two WWTPs

Throughout this study, both plants showed a reduction of >70% in nitrate and nitrite, although WWTP B did not reduce nitrite between September and October (Figure 1). In September and October, WWTPs A and B reduced sulphates and phosphates by 100%, but from June to August, the reduction was less than 50% (Figure 1 and Figure 2). This decrease was brought about by the presence of organic substrate, which is essential for achieving the appropriate treated volume and the required effluent water quality, particularly during these rainy months (September and October) [59].

Additionally, sulphate-reducing bacteria (SRB), the technology utilised in the oxidation pond, may have decreased. Boyd and Mbelu [60] claim that the decrease was caused by an overabundance of biological growth, which prevented the effluent from passing through the oxidation and trickling filters’ inadequately good media. This was brought on by clogged orifices that cause effluent to be distributed unevenly. Phosphate removal bacteria will not be able to survive as a result of the stone media drying out and becoming inoperable [61]. Throughout the investigation, WWTPs A and B experienced COD reductions of 60–90% and 75–90%, respectively (Figure 1 and Figure 2). This is thought to be sufficient as a typical plant’s main treatment (oxidation pond technology) requires a limit of 30 to 40% [60]. For both WWTPs, the turbidity reduction was greater than 50%, with none in August. Although the alkalinity reduction ranged from 0% to 60%, only the months of September (WWTP A) and August (WWTP B) showed an alkalinity drop of roughly 60% (Figure 1 and Figure 2). Temperature, pH, TDSs, EC, Cl^−^, and other parameters were less than 40% for WWTP A and roughly less than 50% for WWTP B. Perhaps because of using the identical oxidation pond technology, WWTPs A and B demonstrated comparable efficacy in terms of pollutant reduction trends. Both WWTPs’ operational capacities showed that they are overburdened and overloaded with sewage wastewater compared to their initial capacity. Studies performed by Qadir et al. [62] and Ouali et al. [63] suggested that the release of unregulated, insufficiently treated, and untreated effluents from wastewater usually introduces hazardous substances into aquatic environments, leading to a decline in water quality to the extent that surface water becomes unsuitable for human consumption and agricultural irrigation. Infrastructure deficiency normally has a greater impact on the efficiency of managing the waste load, which further negatively affects the communities that use the water directly [64]. Lack of clarity regarding the institutional arrangement with dysfunctional coordination between the water bodies and water service providers in terms of unclear policies and implementation has a greater impact on the efficiency of functional wastewater management, as suggested by Haldar et al. [65] and Mabadahanye et al. [66]. Malakane and Maphanga [67] and Burgan et al. [68] further suggested that the integrity of South Africa’s wastewater treatment infrastructure and institutional roles are of paramount importance in protecting the nation’s water resources, environment, and public health. 

### 3.3. Limitations of This Study

In addition to a poorly treated final effluent from wastewater treatment plants that could impact the quality of the river water, reservoirs or other nonpoint sources may be releasing illegal raw sewage directly into the river (upstream of the nearby rivers). These sources were not included in this study. Additionally, this study only considered two WWTPs; adding more will yield more definitive data about the level of pollution caused by the release of final effluents that have not been adequately treated and do not adhere to the established national guidelines or standards. More comprehensive long-term and seasonal studies are required to track the effectiveness of the WWTPs and the possible health risks to the nearby community of Vhembe, as this study was only carried out for five months.

### 3.4. Conclusions

For many South African rural communities, controlling and maintaining the quality of freshwater and sanitation is still a pipe dream. This study examined the upstream and downstream sections of the two WWTPs’ surrounding rivers, as well as the water quality of the influent and effluent. The quality of the discharged final effluent was stated in the findings. Some metrics, such as COD, EC, phosphate, and ammonia, did not meet the established criteria for the bulk of the monitoring period for both WWTPs, but others, such as temperature, pH, TDSs, nitrite/nitrate, turbidity, chloride, alkalinity, sulphate, and free chlorine, did. This study also found that both WWTPs had a higher percentage of nitrate, sulphate, phosphate, and COD contaminants, as well as alkalinity. WWTPs A and B showed minor reductions in temperature, pH, TDSs, EC, and Cl^−^. This demonstrated a general worsening in the physicochemical characteristics of the released wastewater effluents as well as the receiving watershed, implying that the treatment workers are inefficient at producing acceptable-grade effluents, providing an additional environmental health risk. There is also a need for monitoring and prompt action to prevent the indiscriminate pollution of water and the environment, which would reduce the release of improperly treated effluents. Nevertheless, to completely remove or reduce the phosphates, ammonia, sulphates, and nitrates in various zones, an integrated system using modern technologies, like the activated sludge process, advanced oxidation processes, modular water treatment systems, membrane technologies, and emerging technologies like microbial fuel cells and bio-based remediation, must be taken into consideration.

## Figures and Tables

**Figure 1 ijerph-22-00856-f001:**
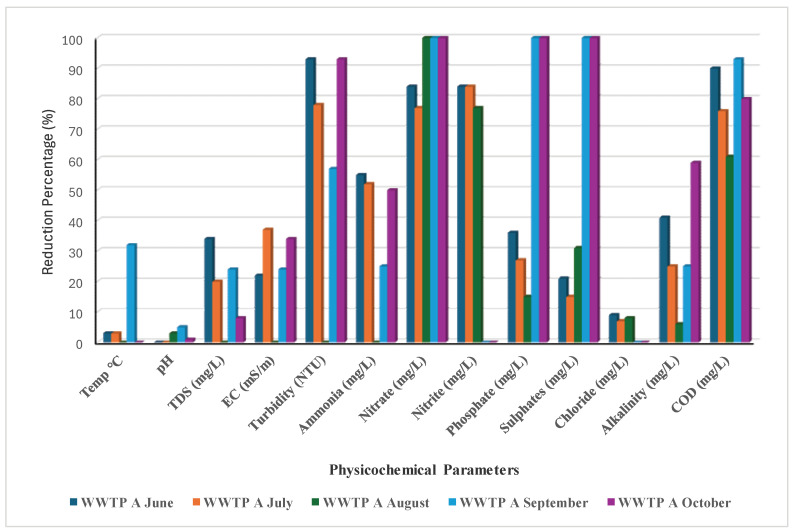
Reduction % of the physicochemical parameters by WWTP A.

**Figure 2 ijerph-22-00856-f002:**
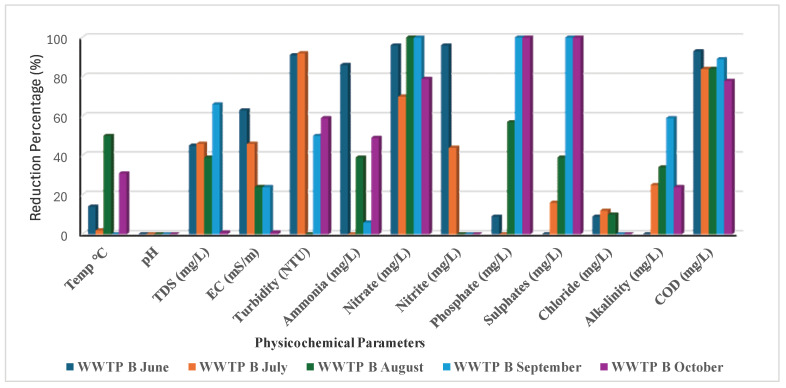
Reduction % of the physicochemical parameters by WWTP B.

## Data Availability

All relevant data are included in this paper and the Appendix A.

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
