# Peer review of "Assessing the Physiochemical Parameters and Reduction Efficiency from Two Typical Wastewater Treatment Plants in the Vhembe District in South Africa"

_ijerph, 2025, doi:10.3390/ijerph22060856_

Round 1
Reviewer 1 Report
Comments and Suggestions for Authors
1. The abstract is overly lengthy, with specific instruments and parameters that could be moved to the materials and methods section rather than being fully presented in the abstract.
2. It is recommended to supplement detailed experimental protocols and instrument principles in Chapter 2, as the current content lacks reproducibility.
3. Conclusions should not be included in the results and discussion section.
4. The article merely examines water quality upstream and downstream of two wastewater treatment plants, lacking innovation. It neither proposes new methods nor demonstrates significant scientific value.
Author Response
- The abstract is overly lengthy, with specific instruments and parameters that could be moved to the materials and methods section rather than being fully presented in the abstract.
Response
Thank you for your comment
This has been amended in the manuscript
- It is recommended to supplement detailed experimental protocols and instrument principles in Chapter 2, as the current content lacks reproducibility.
Response
Thank you for this comment
The protocol has been included in the supplementary material as suggested
- Conclusions should not be included in the results and discussion section.
Response
Thank you for this comment
We have concluded the findings under the conclusion section (lines 397-412) not under the results and discussion section as stated.
- The article merely examines water quality upstream and downstream of two wastewater treatment plants, lacking innovation. It neither proposes new methods nor demonstrates significant scientific value.
Response
Thank you so much for the comment
We stated that the samples analysed were from the influent and effluents as well as upstream and downstream of the adjacent rivers where they discharge their final treated effluents (lines 131-133). This was to investigate whether the final effluent was impacting these rivers negatively or whether the rivers were already impacted before discharging final effluent due to various activities along the rivers especially on the upstream or downstream.

Reviewer 2 Report
Comments and Suggestions for Authors
1. The abstract is too long and requires shortening.
2. The article analyzes the quality of raw and treated wastewater at two wastewater treatment plants (WWTPs). The studies were conducted over a specific period of only 5 months. A proper analysis would be valid for the entire calendar year.
3. The authors should explain the scientific contribution. The introduction does not sufficiently emphasize the novelty and scientific contribution of the presented problem.
4. The capacity of the wastewater treatment plant (WWTP) should be expressed in m3/d.
5. The "Study area description"point lacks a discussion of the technological sequence of wastewater treatment plant. It would be useful to present the technological sequence of wastewater treatment in a block diagram together with the marking of wastewater collection points for analysis.
6. The same information is in lines 126 and 127.
7. The article does not contain any innovative research, calculations or theoretical hypotheses related to the topic.
Author Response
Reviewer Two’s comments
- The abstract is too long and requires shortening.
Response
Thank you for your comment
This has been amended in the manuscript
- The article analyzes the quality of raw and treated wastewater at two wastewater treatment plants (WWTPs). The studies were conducted over a specific period of only 5 months. A proper analysis would be valid for the entire calendar year.
Response
Thank you for your comment
Yes, you are right that we did a snapshot using a grab sample for the study; however, as per the formulated objectives of this study, we wanted to find out if the discharged effluents from two WWTP samples into the adjacent rivers complied with the Department of Water Affairs and Forestry (DWAF) (2010) and (1996) guidelines, the green drop report and their wastewater treatment efficiency as part of the first phase of the study. This was also initiated by complaints from the communities residing in the surrounding areas about the pollution of the rivers. Based on the findings, we have expanded the study over a longer time frame as part of the second phase to include more WWTPs, considering their design capacity, operational and institutional management, and the different technologies employed over a longer period (seasonality) using composite sampling. This will form part of the long-term surveillance study for these WWTPs, and the results will also inform relevant policy development to effectively manage the river catchments in this region. This work will be published by the end of the year.
- The authors should explain the scientific contribution. The introduction does not sufficiently emphasize the novelty and scientific contribution of the presented problem.
Response
Thank you for your comment
This has been amended in the manuscript (lines 93-103).
- The capacity of the wastewater treatment plant (WWTP) should be expressed in m3/d.
Response
Thank you for your comment
This has been amended in the manuscript
However, units of kilolitres per day (kl/d) and cubic meters per day (m³/d) are equivalent when referring to the volume of water. One kilolitre (kl) equals one cubic meter (m³), and since both units are measured per day (d), they represent the same flow rate or volume over 24 hours. Therefore, kl/d and m³/d are interchangeable in measuring water volume.
- The "Study area description"point lacks a discussion of the technological sequence of wastewater treatment plant. It would be useful to present the technological sequence of wastewater treatment in a block diagram together with the marking of wastewater collection points for analysis.
Response
Thank you for your comment
This technological sequence of the two WWTPs has been included in the supplementary material documents 1A and B.
- The same information is in lines 126 and 127.
Response
Thank you for your comment
This information has been included in the supplementary material
- The article does not contain any innovative research, calculations or theoretical hypotheses related to the topic.
Thanks for the comment
Yes, you are right that we did a snapshot using a grab sample for the study; however, as per the formulated objectives of this study, we wanted to find out if the discharged effluents from two WWTP samples into the adjacent rivers complied with the Department of Water Affairs and Forestry (DWAF) (2010) and (1996) guidelines, the green drop report and their wastewater treatment efficiency as part of the first phase of the study. This was also initiated by complaints from the communities residing in the surrounding areas about the pollution of the rivers (lines 93-103). The expanded study that is considering more WWTP facilities visas vis their performance will cover the precise impact of the main parameters influencing water quality. The statistical consideration to either obtain any significance or not, correlations will be established. Based on that study, we are also zooming in on the major parameters that did not comply with South African standards such as such as COD, EC, phosphate, and ammonia. In the ongoing study, we have also considered how the institutional roles of these WWTPs influence their overall effectiveness and performance.

Reviewer 3 Report
Comments and Suggestions for Authors The presented study deals with the issue of the presence of basic pollutants in effluents leaving wastewater treatment plants, here from two provincial sources, namely the so-called peri urban plant and rural plants, which were monitored for a longer period of several months, which brings interesting information regarding the discharged waters. All basic parameters were monitored, i.e. temperature, total dissolved solids TDS, turbidity, chemical oxygen demands COD, dissolve oxygen, free chlorine chlorinde, sulphate, poshphate, ammonium and electrical conductivity. The results were compared with local national regulations, in this case with the South African DWAF guidelines. Most parameters complied with these standards, however, some, such as chlorides and TDS, were higher. Simultaneously, it is not possible to exclude that these substances could have reached drinking water, so there is a potential threat to consumers of these waters. The publication is one of many similar ones that are currently appearing and are locally important because they can alarm local authorities. However, comparison of the obtained information with similar information obtained in different parts of the world cannot be carried out, because the obtained data on the pollution of wastewater effluents from WWTPs depends very much on local conditions; such as local industrialization, applied water treatment technologies, climatic conditions, quality of analytical determinations, etc. For example, a similar, very recent study can be mentioned (Duan Y. et al: Water quality characteristics of municipal wastewater treatment plants and the prospect of reclaimed water utilization in lower-middle income and water-scarce areas: A case study of Puyang. Water Cycle Volume 6, 2025, Pages 61-70, if 4,53). Although this study monitored somewhat fewer indicators at a specific wastewater treatment plant in China (especially COD, NH3-N, total nitrogen, total phosphorus), it also concludes that in each monitoring the results either meet or in which indicators they do not meet local standards, which may be somewhat different in different countries. It is a pity that most similar studies do not monitor the so-called emerging pollutants (pharmaceuticals, antibiotics, hormones, antibiotic-resistant genes, etc.) or heavy metals at all. In conclusion, it can be stated that the presented study brings results that confirm that even in different parts of the world, due to local conditions and applied treatment technology, some basic pollutants in wastewater are not degraded, which can lead to their entry into drinking water, which only confirms that it is necessary to change the current decontamination technologies in wastewater treatment plants worldwide. For this reason, it can be concluded that publications of a similar type do not bring any significantly new scientific knowledge, however they have a kind of warning function that indirectly draws attention to the need to change current treatment methods even in small local treatment plants. I recommend add the proposals for technologies into the conclusion that could remove monitorované unwanted pollutants from wastewater treatment plant effluents. I recommend to publish the manuscript after a minor revision.Author Response
I recommend add the proposals for technologies into the conclusion that could remove monitorované unwanted pollutants from wastewater treatment plant effluents. I recommend publishing the manuscript after a minor revision.
Response
Thank you for your comment and this is insightful information that has enriched this manuscript.
This information regarding technologies has been included in the conclusion section of the manuscript (Lines 414-420).
We have also started working on the emerging pollutants, especially in this region where this kind of study is in the rudimentary stage immediately, we completed this study. This is our new frontier in terms of solution-oriented research.

Round 2
Reviewer 1 Report
Comments and Suggestions for Authors
The authors have revised the article according to the reviewer's comments,and the article can be considered to be published.
Author Response
Thank you and we are really grateful.
Reviewer 2 Report
Comments and Suggestions for Authors
I would like to thank the authors for introducing changes to the article in accordance with the comments. However, in my opinion, the content of the article should be supplemented with the technological diagrams of the WWTP A and WWTP B presented in the supplementary material.
Author Response
I would like to thank the authors for introducing changes to the article in accordance with the comments. However, in my opinion, the content of the article should be supplemented with the technological diagrams of the WWTP A and WWTP B presented in the supplementary material.
Response
Thank you for your comment and this is insightful information that has enriched this manuscript.
This information regarding supplementing the content from the supplementary has been incorporated into the manuscript under the description of the study section (Lines 116-121).
